# Corrosion Behavior in Saline Solution of Electrodeposited Nanocomposite Zn-CeO₂ Coatings Deposited onto Low Alloyed Steel

**Loïc Exbrayat** [1], **Christelle Rébéré** [2], **Rémy Milet** [3], **Emilie Calvié** [4], **Philippe Steyer** [5] and **Juan Creus** [2,*]

1    SafranTech, Rue des Jeunes Bois, 78772 Magny Les Hameaux, France
2    LaSIE UMR 7356 CNRS, La Rochelle Université, Av. Michel Crépeau, 17042 La Rochelle, France
3    LaSALYS, 1 av. du Champ de Mars, 45100 Orléans, France
4    Ahlstrom Research Center, 38140 Apprieu, France; emilie.calvie@ahlstrom.com
5    MATEIS, INSA Lyon, 21 av. Jean Capelle, 69621 Villeurbanne cedex, France
*    Correspondence: juan.creus@univ-lr.fr

**Abstract:** Zn-CeO₂ nanocomposite coatings were deposited onto mild steel substrates by electrodeposition process. Our study highlights the effect of ceria nanoparticles embedded into a metallic matrix on the corrosion behavior in saline environment. The experimental results show that the ceria incorporation and dispersion depend on the particles concentration in the electrolyte. High concentrations of particles favor agglomeration and adsorption of agglomerates on the surface of the zinc coating. A slight improvement of the corrosion resistance compared to pure electrodeposited zinc coatings is observed. The beneficial effect seems to be dependent on the dispersion of the nanoparticles embedded inside the mela matrix. The distribution of nanoparticles seems to be the key-parameter influencing the corrosion behavior, permitting to improve the corrosion behavior during extended immersion test.

**Keywords:** metal matrix composites; zinc; electrodeposition; corrosion; EIS





## 1. Introduction

Electroplated zinc and zinc alloy coatings are widely used as sacrificial coatings for the protection of steel structures [1] due to their sacrificial behavior and their low cost. Nevertheless, a major drawback of these coatings is the important release of metallic cations towards the environment resulting from the coating dissolution [2,3]. Challenge in the durability of systems is then twofold: (i) to reduce the dissolution rate of the sacrificial layer on the one hand, (ii) to improve its mechanical properties on the other hand.

A decrease of the zinc reactivity was obtained after the modification of the chemistry of the coating [4]. Addition of alloying elements such as Mg, Mn, Ni, Fe and Co were investigated in the literature and denoted improvement of the corrosion resistance by reducing the heavy metal release towards the environment [5–10]. An alternative may also be based on the incorporation of micro- or nano-particles to the electrolytic bath during the coating's growth to form a so-called metal matrix composite (MMC) coating [11]. The properties of the composite coatings depend on the matrix nature, the amount and distribution of the co-deposited particles. The latter characteristic is related to electrodeposition parameters including particle characteristics (particle shape and size, concentration in electrolyte, surface charge), electrolyte composition (surfactant, additives, concentration), pH and applied current (direct od pulse plating) [12].

Different particles were investigated in the literature, such as TiO₂ Yttria Stabilized Zirconia SiO₂, ZrO₂, SiC, MoS₂, graphene, cotton nanocrystals or PMMA [13–23]. In the literature, it is reported that the increase of the particle concentration in the deposition bath significantly affect the microstructure of the zinc coatings [24]. The particle incorporation

results in a grain refinement with the assumption that they act as heterogeneous sites for zinc nucleation [12]. Kumar et al. observed that the incorporation of $WO_3$ nanoparticles leads to more uniform, compact and smooth deposit and a slight refinement was observed when the nanoparticle concentration in the bath is increased [25–36]. Anwar et al. suggest that the adsorption of $TiO_2$ nanoparticles on the surface of the growing Zn-Ni matrix inhibit the deposition process, leading to a refinement of the microstructure [26]. The incorporation of nanoparticles leads to a decrease of the thickness of the metal matrix suggesting that the faradic efficiency is changed due to the inhibition effect of the embedded particles and the variation of the hydrodynamic behavior of the electrolyte [12,26,27]. Microstructure modifications, as grain refinement or evolution of the grain shape, are not systematically observed in all MMC configurations due to the balance between several parameters like nanoparticle size, pH, current density, magnetic or ultrasonic agitation [27–30]. However, authors are quite unanimous in highlighting the significant effect of the embedded particles for the elimination of surface defects, porosities or cracks that affect the surface reactivity. As mentioned in the review of Safavi et al., the critical parameter to achieve a uniform dispersion of particle inside the metal matrix is the particle concentration in the electrolyte. Increasing this concentration may favor the formation of agglomerates lowering the homogeneity of the coatings [27]. Several authors have reported a threshold electrolyte concentration that leads to the formation of agglomerated on the surface of the deposit [26,27,29].

Corrosion studies were based on electrochemical stationary [31] or electrochemical impedance spectroscopy measurements [32]. Salt spray experiments according to the standard ASTM B117 [33] are rarely presented, although these results are of prime interest for a potential industrial scale-up. Most of the authors conclude that the composite structure leads to a better corrosion behavior, but the reason of the improvement is often contradictory. For instance, some studies report that composite coatings display more negative open circuit potential (OCP) values that those of pure electrodeposited zinc coatings [6,34]. This evolution was attributed to an inhibiting character affecting the dioxygen reduction reaction. In opposite, other results attest a nobler behavior for composites compared with the single pure zinc [22,23]. This behavior is link to the nature of the nanoparticles that are able to slow down the kinetics of the zinc dissolution [35,36]. The assumption of a "barrier effect" due to the particles which would protect the metal matrix is also mentioned, but this assumption still remains unclear and often was suggested in nickel bases composites configurations [36,37].

Improvement of the corrosion resistance was associated to the microstructural evolution of the zinc coating. The incorporation of ceramic particles often leads to microstructural or metallurgical modifications of the metal matrix as grain refinement, grain shape evolution or texture evolution. It is difficult to underline the influence of the embedded particle on the corrosion behavior without taking into account all the metallurgical changes. So, we have selected a pure metal matrix with a rapid deposition kinetic in order to limit the evolution of the microstructure or metallurgical features during the co-deposition.

We propose an approach combining a thorough metallurgical characterisation with multi-scale corrosion tests in order to understand the influence of ceria nanoparticles incorporation and dispersion on the corrosion behavior of electrodeposited zinc coatings.

## 2. Experimental

### 2.1. Materials and Coating Elaboration

Cylindrical AISI 4340 steel substrate was embedded in epoxy resin. Composition of the steel substrate is described in [35]. Coatings were obtained through an electrodeposition process achieved in an alkaline bath containing different ceria nanoparticles concentrations. The composition of the zinc electrolytic bath is presented in Table 1.

**Table 1.** Zinc electrolytic bath composition.

| Compound | Massic Concentration (g L$^{-1}$) | Molar Concentration (mol L$^{-1}$) |
|---|---|---|
| ZnCl$_2$ | 63 | 0.45 |
| NH$_4$Cl | 40 | 1.5 |
| SDS (anionic surfactant) | 0.5 | $1.7 \times 10^{-3}$ |
| CeO$_2$ nanoparticles | 0–25 | 0–0.15 |

CeO$_2$ nanoparticles from Sigma Aldrich supplier were added to the plating bath and a continuous stirring during 12 h was performed to ensure a dispersion of nanoparticles into the plating bath. The pH was then adjusted to 8.9 $\pm$ 0.1 by addition of a 8 mol L$^{-1}$ NH$_4$OH solution. Sodium DodecylSulfate (SDS from Sigma Aldrich$^{\circledR}$) was used as anionic surfactant.

Ceria nanoparticles had a size ranging from 15 to 25 nm. Coatings were deposited in direct current mode at a fixed current density of 30 mA.cm$^{-2}$ using a conventional 3-electrodes cell with a zinc anode and a saturated calomel reference (SCE) inserted in a Luggin capillary. The electrolytic solution was stirred and temperature was controlled at 30 °C using a cryothermostat. Three different concentrations of ceria nanoparticles in the electrolytic bath were tested and compared: 0 g L$^{-1}$ (pure zinc coating), 15 g L$^{-1}$ and 25 g L$^{-1}$. Gravimetric measurements were made in order to deduce the cathodic efficiency that is about 74, 40 and 31% respectively for pure zinc, 15 g L$^{-1}$ and 25 g L$^{-1}$ ceria. Pure zinc electrodeposited coating thickness is approximately 8 μm, so the duration of the galvanostatic tests for the composite coatings was adjusted in order to obtain similar coating thickness.

In order to check the reproducibility, several coatings were prepared, their morphological and chemical composition being systematically compared [38].

*2.2. Coating Characterization*

The coating surface was examined by Scanning Electron Microscopy (SEM, Zeiss SUPRA 55 VP, from ZEISS company, France). The cross section of the coating was also observed using a FiB-SEM Zeiss dual FIB-FEG nVision40, using a very low current for the gallium beam in order to avoid interactions between the Zn, Fe and Ga elements. SIMS analyses were performed using a CAMECA IMS 7F, from AMETEK france with a tension of 5 V and an incidence angle for the Cs ion beam of 44°.

X-ray diffraction (XRD) analyses were also performed before and after salt spray test in order to investigate corrosion products formation. A Bruker Advance D8 diffractometer equipped with a Cu radiation was used adopting the following parameters: step size of 0.03°, scan step time of 3 s in the [25–96°] range from before corrosion test and in the [5–96°] range after corrosion to detect all the potential zinc-based corrosion products.

*2.3. Corrosion Behavior in Saline Solution*

The main electrochemical experiments were performed in a conventional three electrode glass cell using samples as working electrode and connected to a Pekin Elmer EG&G 273A potentiostat from HTDS company. The counter electrode is a large platinum grid. The potentiodynamic experiments were achieved after 1 h of immersion in saline solution. Before each electrochemical experiment, surfaces of the samples were delimited using a Lacomit$^{\circledR}$ varnish in order to keep the surface constant at 1 cm$^2$. Potentiodynamic polarization curves were plotted from $-150$ mV (vs OCP) in the cathodic side up to $+150$ mV in the anodic side using a sweep rate of 0.2 mV/s. The corrosion potential $E_{corr}$, current density $j_{corr}$ and the anodic Tafel slope $b_A$ were estimated using the Tafel extrapolation [39]. The polarization resistance was calculated from the relation (1) considering that the cathodic

reaction in aerated saline solution is mainly controlled by the diffusion of dioxygen species toward the electrode surface, leading to $b_C >> b_A$.

$$R_p = \frac{b_A b_C}{2.3(b_A + b_C)} \cdot \frac{1}{j_{corr}} = \frac{b_A}{2.3 j_{corr}} \tag{1}$$

To check reproducibility, at least two identical experiments were done for each configuration.

Durability of samples was investigated through different complementary corrosion tests: extended immersion tests of 96 h in saline solution combining OCP evolution and polarization resistance Rp measurements and electrochemical impedance spectroscopy (EIS) measurements during extended immersion tests. Immersion test consists in measuring the open circuit potential (OCP) of coated samples as a function of immersion time in a 35 g·L$^{-1}$ saline solution at 25 °C. Polarization resistances were measured at different immersion times by sweeping the potential at ± 20 mV around the OCP with a sweeping rate of 0.2 mV/s.

Electrochemical Impedance measurements were performed with a Metrohm Autolab PGSTAT 302N potentiostat, from Metrohm France manufacturer at the OCP during a 96 h immersion test in the same conditions as exposed previously for the measurement of the polarization resistance. Impedance spectra were acquired in the frequency range from 100 kHz to 0.5 Hz with an AC voltage amplitude of 10 mV around the OCP. 8 points per decade were recorded. Impedance spectra were then analyzed using an equivalent circuit model fitted on Zview® 2.3d software. Figure 1 describe the Equivalent electric circuit model used for the fitting of the corrosion behavior of the zinc-based coatings during extended immersion test.

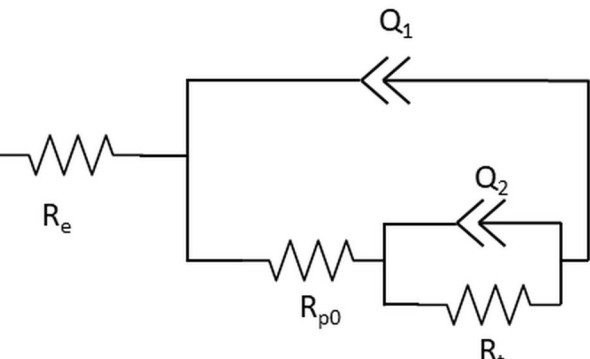

**Figure 1.** EIS equivalent circuit model for the fitting of the electrochemical behavior of the composition coatings during extended immersion test.

The equivalent electric circuit model is typical the model used for porous electrodes composed of the electrolyte resistance Re and 2 parallel circuits (RpoQ1) and (RtQ2) respectively characteristic of the surface degradation and formation of corrosion product film. CPE were considered due to the surface charge distribution and the capacitance was estimated from the Brug equation and the discussion of Hsu et al. [40,41].

### 2.4. Neutral Salt Spray

In order to evaluate the protective efficiency of coatings at the industrial scale, neutral salt spray test was performed using an Ascott CC450XP accelerated cyclic corrosion chamber, from Labomat manufacturer France. The experiment was carried out according to the ASTM B117 standard that consists of a continuous spraying of a 50 g L$^{-1}$ NaCl saline solution at a temperature of 35 ± 1 °C. Unconsidered edges of the samples were covered with Lacomit® varnish to avoid crevice corrosion. For each composition of samples, three identical specimens are exposed for a reproducibility purpose.

## 3. Results and Discussion

### 3.1. Metallurgical Study

Metallurgical states of the zinc and zinc composite coatings were investigated before corrosion experiments. SEM surface observations of the electroplated samples are presented in Figure 2. Zinc coatings morphology corresponds to typical packets of hexagonal platelets with different orientations, stacked on each other. The size of the platelets is around few micrometers but their thickness is very thin, close to few ten nanometers. Figure 2 confirms the incorporation of ceria nanoparticles when they are present in the deposition bath. The higher the concentration of ceria nanoparticles in the bath, the higher the particle amount is incorporated into the coating as reported in literature [25–30]. But for high particle concentration in the electrolyte, adsorption of agglomerates on the zinc coating surface is favored [26,27]. As reported in [35], addition of ceria nanoparticles does not strongly influence the zinc morphology. Hexagonal platelets are still observed and it seems that the orientation of the platelets is slightly influence by the presence of particles in solution. When the ceria concentration is increase, the zinc composite coatings seem to be denser and smoother as suggested by Lopes et al. [23]. High magnification reveals the presence of agglomerated nanoparticles adsorbed on the zinc surface, the highest density is reported for the 25 g L$^{-1}$ samples.

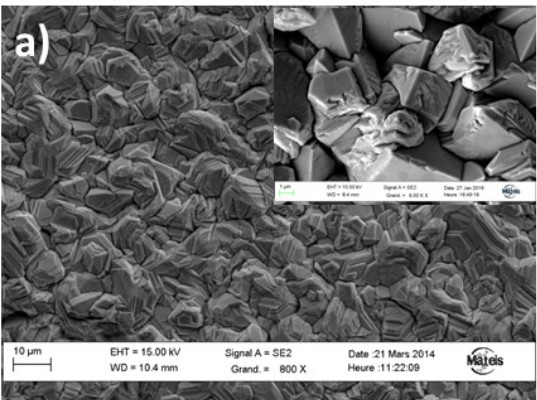
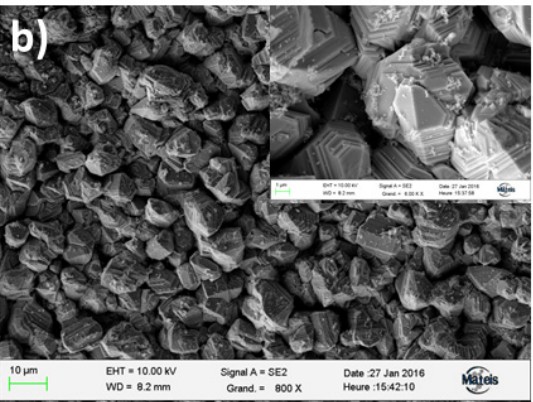
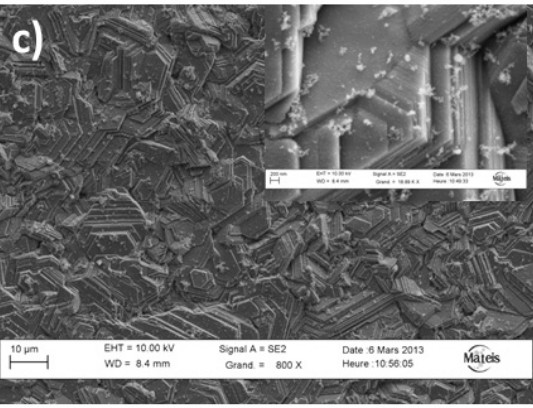

**Figure 2.** Surface SEM images at two different magnifications of (**a**) Zn-CeO$_2$ 0 g L$^{-1}$; (**b**) Zn-CeO$_2$ 15 g L$^{-1}$; (**c**) Zn-CeO$_2$ 25 g L$^{-1}$.

XRD diffractograms presented in Figure 3 reveal that the diffraction peaks of the MMC coatings are similar to that of pure electrodeposited zinc coating. The main diffraction peaks of pure zinc at 36.4 and 43.5° are observed with an additional diffraction peak close to 28.6° corresponding to the (111) diffraction peak of CeO$_2$ is detected at high ceria concentration in solution. A slight difference of texture may be detected with the Zn(002) peaks at around 36.4° that is more intense for coatings containing particles and may explain the slight

difference in morphology. Such stronger intensity could be due to a more pronounced 2D character of grains, as also indicated by the more angular character morphology of the surface (Figure 2b,c vs. Figure 2a). The increase of the fraction of planes with higher packing density like the Zn(002) planes, favors the corrosion resistance due to more compact surface planes and less surface free energy [23,42].

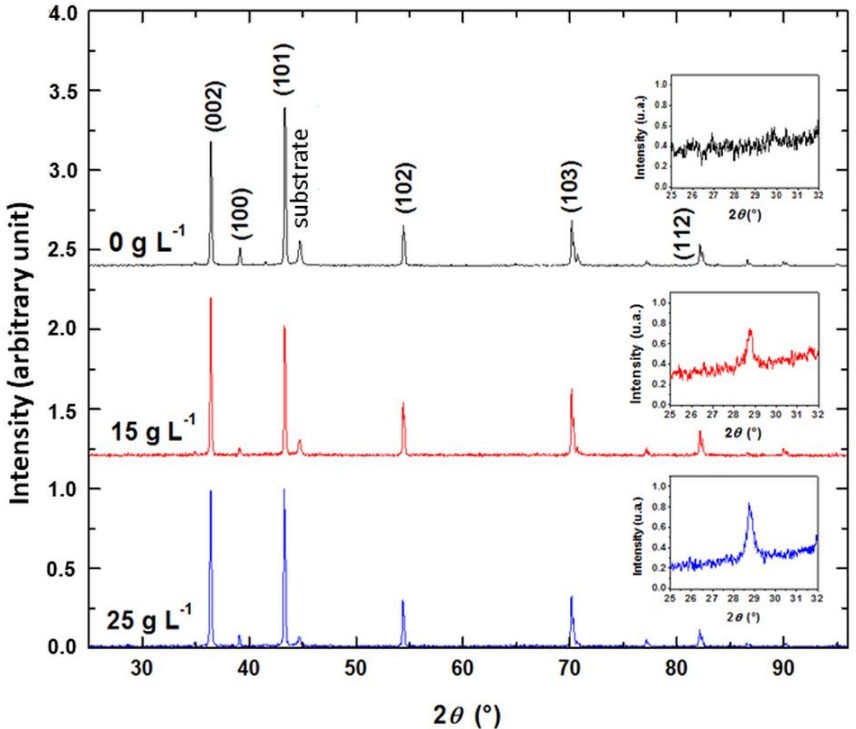

**Figure 3.** XRD diagrams of electrodeposited Zn based coatings onto steel substrate.

This micro-granular microstructure is typical of zinc coatings deposited from conventional electrolytic baths without additives [43–45]. Particles seem to be preferentially adsorbed at the edges of zinc platelets during their lateral growth [38,46] as it can be seen in Figure 2b,c.

SEM cross sections were performed and the results confirm that the microstructure is not changed by the incorporation of ceria nanoparticles. FIB-SEM cross section of the Zn-CeO$_2$ 25 g L$^{-1}$ coating is presented in Figure 4. A dense and columnar structure is observed. The width of the columns corresponds to the dimension of the hexagonal platelets detected on the SEM surface observations, around few micrometers. The average thickness is around 6–7 µm, but the roughness is quite important due to the angular morphology more pronounced for this coating. Ceria nanoparticles were mainly detected on the surface of the coatings. EDS maps were performed on surfaces and along the cross section of the coated samples and it has been observed that the increase of the amount of ceria nanoparticles into the electrolytic bath favors higher detection of cerium inside the zinc metallic coating.

When the particle concentration in the electrolyte increase, ceria nanoparticles are mainly located at the coating surface. This location is due to the agglomeration of nanoparticles in solution due to the modification of the electrolyte characteristics. GDOES and SIMS analyses were performed on the MMC. SIMS element maps for Ce, Zn and Fe elements, presented in Figure 5, permit to highlight the dispersion of ceria particles into the metallic matrix. This analysis underlines the presence of an optimum concentration of ceria nanoparticles in the electrolytic bath that favors a quite homogeneous dispersion of ceria nanoparticles inside the metallic matrix. For the 25 g·L$^{-1}$ configuration, the amount of incorporated ceria particles determined by EDS or XRF analyses is more important compared to 15 g L$^{-1}$ because agglomeration is emphasized and ceria particles are essentially located at the coating surface [35].

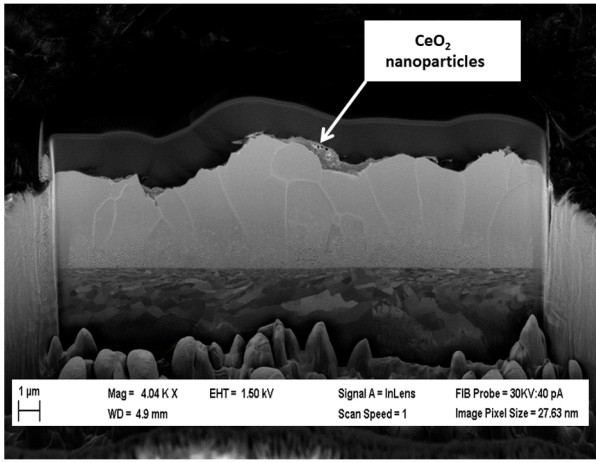

**Figure 4.** FIB-SEM cross section of the electrodeposited Zn-CeO$_2$ 25 g·L$^{-1}$ coating.

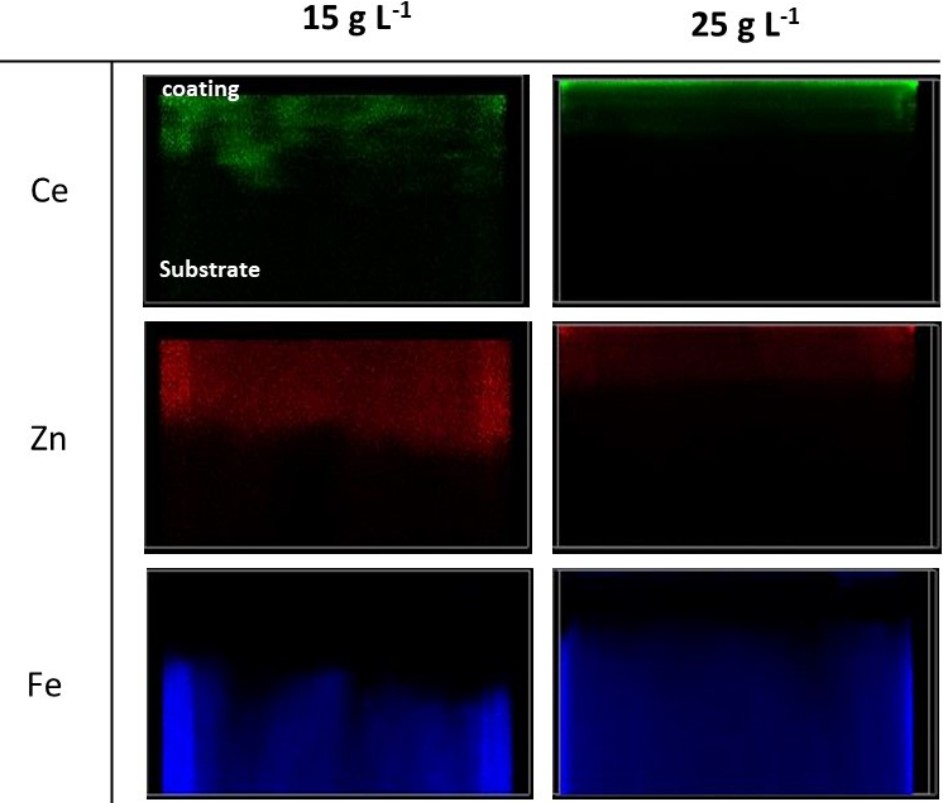

**Figure 5.** SIMS elements cartography in composite based Zinc coatings.

Ceria nanoparticles are mainly adsorbed at the surface of the Zn-CeO$_2$ 25 g·L$^{-1}$ coating that could be favored by the angular shape morphology of the Zn platelets. GDOES profile, not presented, also suggest a slight increase of the ceria nanoparticles at the coating/substrate interface, perhaps due to the presence of an important density of defects that would favor the trapping of the nanoparticles, for example the presence of pores or cavities.

The embedded particles inside the metal matrix are located inside small cavities mostly located along grain boundaries, as presented in [38]. The cavities can play the role of hosting sites where nanoparticles could be entrapped during the zinc electrodeposition process [23,38]. Holes containing nanoparticles are quite isolated in the coating thickness due to large zinc grain sizes (a few micrometers). The particle incorporation permits to limit the presence of defects. For the 15 g L$^{-1}$ sample, the nanoparticles are introduced

into the coating along the grain boundaries or close to the substrate/coating interface, that could favor the densification of the coating [26].

### 3.2. Corrosion Properties of Nanocomposites Zn-Ceria Coatings

In order to correlate the corrosion resistance of pure and nanocomposite zinc coatings with their metallurgical state and particle distribution, corrosion tests were performed on the three configurations of coating with a quite similar thickness (around 8 μm).

#### 3.2.1. Polarization Curves after One Hour of Immersion in Saline Solution

Polarization curves obtained after one hour of immersion in a 35 g L$^{-1}$ saline solution of pure zinc and ceria composite coatings are presented in Figure 6. All the electrochemical parameters were obtained according to the Tafel extrapolation and the values are summarized in Table 2 [47].

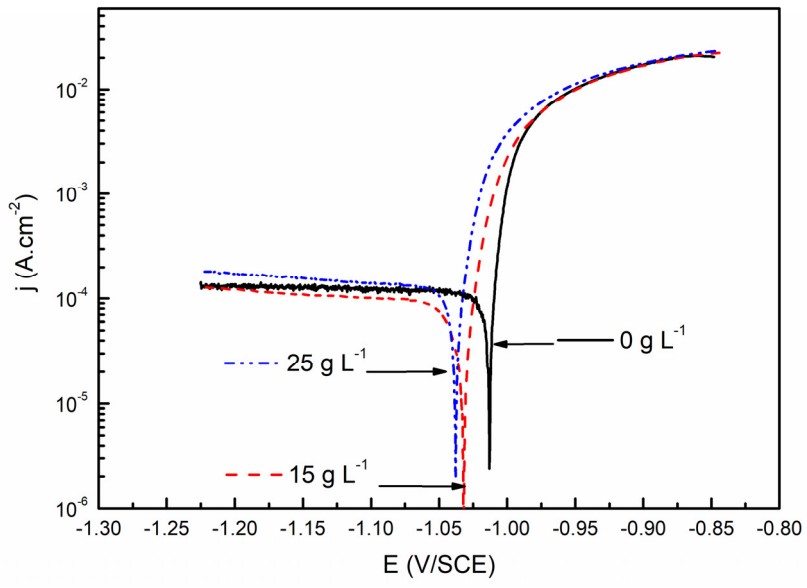

**Figure 6.** Polarization curves after 1 h of immersion in 35 g·L$^{-1}$ saline solution.

**Table 2.** Electrochemical parameters deduced from the polarization curves after 1 h of immersion in saline solution.

| Nanoparticle Concentration in the Electrolytic Bath | $E_{corr}$ (V/SCE) | $j_{corr}$ (μA/cm$^2$) | $b_A$ (mV/dec) | $b_C$ (V/dec) | Rp (Ω.cm$^2$) |
|---|---|---|---|---|---|
| 0 g L$^{-1}$ | −1.013 | 100 | 15 | −3.2 | 65 |
| 15 g L$^{-1}$ | −1.032 | 90 | 25 | −2.4 | 122 |
| 25 g L$^{-1}$ | −1.038 | 120 | 21 | −1.6 | 76 |

The cathodic and anodic Tafel slopes $b_C$ and $b_A$ are reported in Table 2 for pure zinc and MMC coatings during an immersion of 1 h in saline solution. For pure zinc electrodeposited coating, the Tafel slopes are in good agreement with the values reported in literature [10,48]. For pure zinc coating, the high value of the cathodic slope $b_C$ suggests a reduction reaction controlled by the dioxygen diffusion towards the zinc surface. For the MMC samples, $b_C$ decreases with the amount of ceria nanoparticles in the deposition solution. This evolution suggests an additional contribution in the cathodic reaction of the adsorbed ceria agglomerate particles. The dioxygen reduction reaction also occurs on the surface of these agglomerates particles, explaining a mixed contribution of activation-diffusion in the cathodic reaction. Concerning the anodic Tafel slopes, the values evolve between 13 to 37 mV/dec and are in good agreement with the literature [10]. A slight

evolution of the anodic Tafel slope is observed, suggesting that the ceria nanoparticles are also involved in the dissolution mechanism of zinc in saline solution.

Polarization resistance Rp after one hour of immersion in saline solution is calculated using Equation (1). Rp increases with the incorporation of ceria nanoparticles, but at 25 g L$^{-1}$, this increase is reduced due to the heterogeneous distribution of the nanoparticles inside the zinc coating and the presence of large agglomerates on the coating surface. When the concentration of particles into the electrolytic deposition bath is high, agglomeration of particles is favored and the agglomerates are mainly located at the coating surface. It was reported that this preferential distribution of the agglomerates on the top surface decreases the corrosion resistance afforded by the particles. Localized degradation was suggested to occur beneath or at the junction of the agglomerates, accelerating the dissolution of the metallic matrix [26–30].

In the cathodic domain, values of the current densities remain quite constant on large cathodic overpotentials meaning that the cathodic kinetic is mainly controlled by a pure diffusion step, which is the kinetic limiting process. Experiments are carried out in stirred and aerated solutions, so the cathodic reaction is mainly associated to the dioxygen reduction reaction on the zinc matrix surface. On nanocomposite zinc coatings, an evolution of the slope of the cathodic branch is observed. This evolution could be associated to the presence of ceria nanoparticles on the coating surface that progressively modify the rate determining step in the global cathodic reaction of dioxygen reduction. According to the literature [49,50], ceria particles at the nanometric scale is able to storage oxygen and support the cathodic reaction. This implies a modification of the reactivity of the MMC surface.

The 25 g L$^{-1}$ sample presents a higher amount of ceria nanoparticles on top surface compared to the 15 g L$^{-1}$ one. Effectively, SIMS analyses underline that an important part of the nanoparticles are agglomerated suggesting that the distribution of these particles on the coating surface is not uniform limiting their inhibition effects. In opposite, for the 15 g L$^{-1}$ sample, a homogeneous dispersion was observed on the surface, which may explain the lower values of the cathodic current densities. Moreover, a too high amount of ceria is not recommended since it has been reported that CeO$_2$ nanoparticles were able to adsorb the dioxygen on their surface [51], contributing to the cathodic reaction. A compromise has then to be found between the inhibiting influence of ceria and its oxygen reservoir effect. The novelty of our research is focused on the fact that the 15 g L$^{-1}$ concentration presents a good balance between the reactivity increase by the dioxygen storage behavior of the ceria nanoparticles and the inhibition effect due to the surface coverage.

The incorporation of nanoparticles leads to a slight shift of the corrosion potential towards more negative values [52]. The shift of the corrosion potentials is in good agreement with the study of Ranganatha et al. on the Zn-CeO$_2$ system [34]. These coatings were synthesized using a CTAB cationic surfactant which favors acicular morphology with zinc plates perpendicular to the substrate, compared to our results using SDS anionic surfactant that does not affect the electroplated zinc morphology (Figure 2) [27]. Ranganatha et al. had also reported a shift of the corrosion potential towards more negative values (−1.08 V/SCE) compared to zinc coating (−1.03 V/SCE) in saline solution. He suggested that this evolution was linked to the localization of the embedded nanoparticles at grain boundaries (GB). Indeed, incorporation of nanoparticles could modify the nature of the GB, which may influence the corrosion kinetics [53]. Other authors reported similar trends in the literature with electrodeposited zinc composite coatings [6,54]. Vlasa suggested that TiO$_2$ particles present on the coating surface acted as uniform passive sites reducing the degradation of the coating [54]. Frade et al., studying the Zn-TiO$_2$ system, also observed a slight shift of the potential towards negative values compared to pure zinc coating. Anyway, potential differences are very low (less than 30 mV), suggesting a progressive loose of the inhibition effect due to the adsorbed particles on the surface of the coatings. The oxygen storage ability of ceria particles is mainly associated to the creation of oxygen vacancies at the surface of ceria nanoparticles. According to the works of Sayle et al. using atom-level

modelling to describe the surface activity of ceria particles for catalyst applications, it was reported that oxygen depletion ability was strongly dependent on the shape, size and defect contents of the nanoparticles [55,56]. Our commercial ceria nanoparticles with an average size around 25 nm present a polyhedral morphology composed of misoriented grains separated by grains boundaries and triple junctions as presented in [39]. These morphological parameters could affect the oxygen adsorption on the nanoparticle surface.

### 3.2.2. OCP and Rp Evolutions during Extended Immersion Tests

Open circuit potentials (OCP) and polarisation resistance (Rp) were measured during a 96 h immersion test in a 35 g $L^{-1}$ NaCl solution (Figure 7). Reproducibility was performed for each test and same evolutions were obtained for the electrochemical parameters.

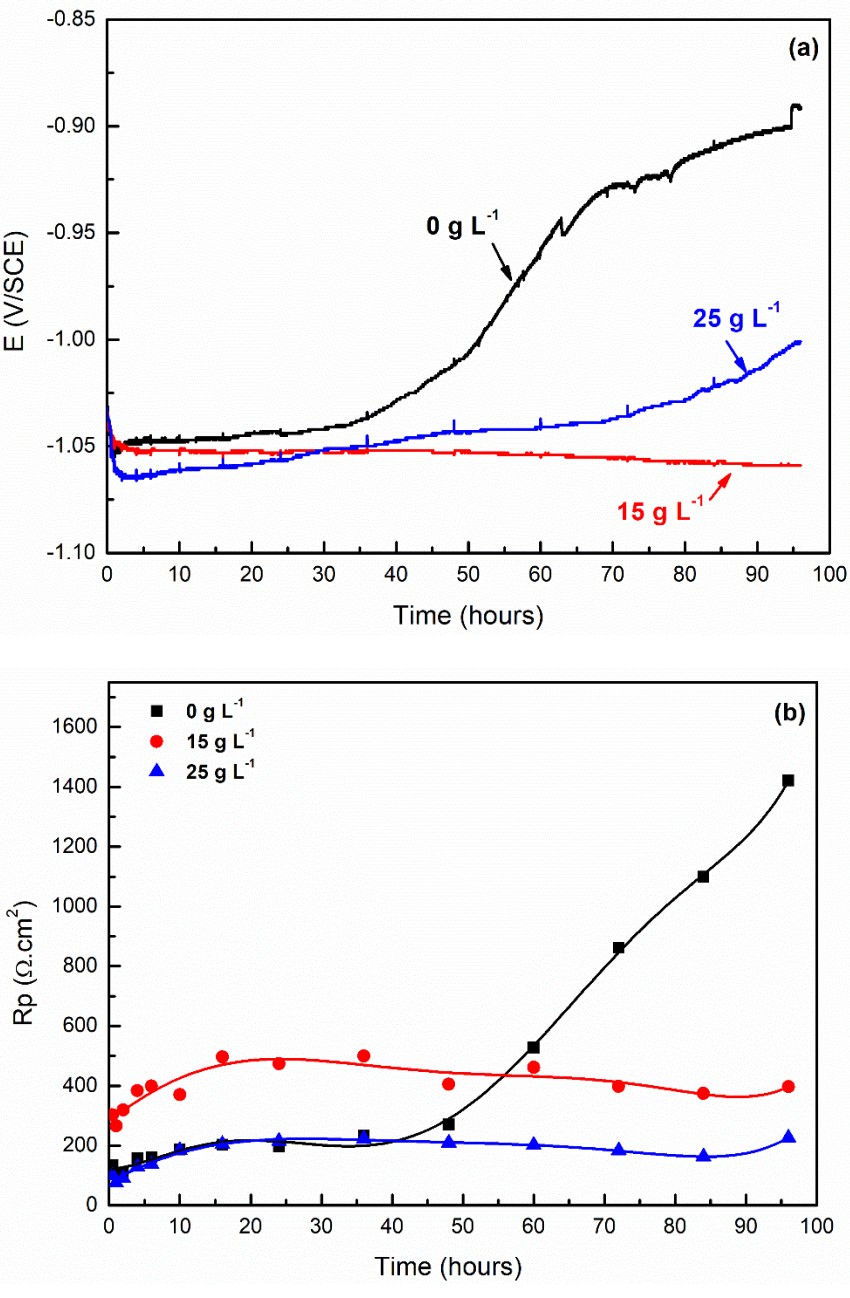

**Figure 7.** OCP (**a**) and $R_P$ (**b**) evolutions versus time for pure zinc, 15 g $L^{-1}$ and 25 g $L^{-1}$ Zn-$CeO_2$ composite coatings electrodeposited on steel substrate, immersed during 96 h in a 35 g $L^{-1}$ saline solution.

The open circuit potentials are in the same order of magnitude at the beginning of the immersion test. White rust is rapidly observed for all samples few minutes after the beginning of the immersion. OCP of the nanocomposite Zn coatings evolves slightly towards more negative potential during the first hours of immersion. This could be associated to the preliminary dissolution of the native "zinc oxide" layer initially formed in contact with air and not protected by the adsorbed $CeO_2$ particles in Zn surface. OCP is then stabilized. It corresponds to the coverage of the coatings by zinc based corrosion products that probably act as diffusion barrier. The corrosion products limit the access of oxygen limiting the corrosion rate. These compounds are mainly composed of zinc corrosion products with low solubility and they precipitate on the surface of the coating [57].

Ennoblement of the open circuit potential happens at around 35 h of immersion for pure zinc coating. This ennoblement is associated to the degradation of the zinc layer. We can suppose that the progressive consumption of the zinc coating progressively permits to reveal open porosities, so we can assume an important contribution of the steel substrate exposed to the saline solution. OCP values of the Zn coated sample slowly converge towards that of the steel substrate covered by a porous corrosion product film. This evolution is significant with a progressive loss of the sacrificial behavior of the coating as observed in [58]. For the 25 g $L^{-1}$ composite coating, OCP remains quite constant and an ennoblement of the potential happens after 85 h of immersion suggesting the beginning of the loss of the sacrificial behavior. The incorporation of the $CeO_2$ nanoparticles with a heterogenous distribution improves the durability of the coating.

For the 15 g $L^{-1}$ coating, the open circuit potential remains quite constant throughout the immersion test. As the thicknesses and morphologies are quite similar for all the coatings, the distribution of the particles on the surface and inside the metallic matrix would be at the origin of the better corrosion behavior of the 15 g $L^{-1}$ Zn composite coatings in saline solution.

These results are consistent with those reported by Xia et al. [7]. Presence of yttria nanoparticles could play the role of a physical barrier that might reduce the active surface of the coating in contact with the corrosive solution. Therefore, the second ceramic phase could slow down the zinc dissolution rate. Moreover, particles could also influence the mechanical anchorage of the corrosion products film, which would lead to a better protection.

Similar observations were reported by Azizi et al. in the case of Zn-$SiO_2$ composite coating systems during a 14 days immersion test in saline solution. Compared to pure zinc coating, the open circuit potential of the composite coating progressively became nobler than pure zinc coatings due to the presence of the corrosion product film [31]. Similar analyses were suggested by Lopes et al. on the corrosion resistance of Zinc-cotton nanocrystal composite coatings. It was suggested that the particles, agglomerated on surface, favor the formation of the corrosion products that act as a protective barrier against corrosion [23].

We could also suggest, in order to explain the slight difference after 80 h between the 15 and 25 g $L^{-1}$ configurations, that an excess of particles could also induce more defects in the coating, affecting its long-term protection.

Rp values at the beginning of immersion are relatively low (around 170 $\Omega.cm^2$) for the pure zinc and 25 g $L^{-1}$ coatings compared to the 15 g $L^{-1}$ that presents a Rp value close to 250 $\Omega.cm^2$. The Rp value of the pure zinc coating is in good agreement with the values described in literature for the platelet-shape morphology that presents a very low compactness [59,60].

Distribution of ceria nanoparticles on the coating surface has an important role during the first stage of the coating degradation mechanism. It is reported in literature that the oxygen storage capacity of the ceria nanoparticles permits to affect their surface reactivity. The oxygen vacancies at the surface of the NPs are balanced by the reduction of Ce(IV) in Ce(III) [56]. So, at the beginning of the immersion, a compensatory effect was noticed for the composite coatings between the reduction of the active surface of the coating and the increase of the cathodic current density due to the reactivity of the ceria particles.

After few hours of immersion, polarization resistance increases slightly. This increase could be due to the formation of adherent corrosion products that limit the diffusion of aggressive species such as oxygen and chlorides towards the metal/solution interface. Then, a stabilization of the $R_p$ is observed and the Rp values remain quite constant during the immersion except for the deposit Zn-CeO$_2$ 25 g L$^{-1}$ for which a slight ennoblement is noticed after 80 h. The amplification of the surface reactivity due to the presence of the ceria particle was rarely discussed in the literature; the particles are supposed to be electrochemically inert or micro-galvanic cells were reported between the particles and the metal matrix that suppress the localized degradation of zinc matrix [34,61]. In our study, micro-galvanic cells are detected between adsorbed ceria nanoparticles at the surface and zinc matrix that favors the formation of adherent and dense corrosion products.

For the pure zinc coating, polarization resistance values increase continuously after 40 h of immersion, in parallel with a progressive ennoblement of the open circuit potential. This potential evolution could be due to the progressive loss of the sacrificial protection of the zinc coating. In opposite, for composite coatings, $R_p$ values and open circuit potential are quite stable throughout the immersion test, suggesting that the composite coatings still offer sacrificial protection to the steel substrate. This specular behavior is directly dependent on the presence and distribution of the ceria particles. According to the literature, the oxygen vacancies formation in the ceria particles leads to the reduction of Ce(IV) in Ce(III) [55,56]. Presence of Ce(III) species is known to play an inhibitive role preventing the zinc oxidation as reported by Poupard et al. [62,63]. Furthermore, Rp values are higher for the 15 g L$^{-1}$ compared to that of the 25 g L$^{-1}$, this difference being mainly associated to the nanoparticles distribution on the coating surface. A homogeneous distribution should favor the formation of a more compact or adherent corrosion product films as suggested by Hashimoto et al. [64] and the electrochemical reactions at the surface of the ceria particles permits to inhibit the zinc dissolution.

### 3.2.3. Electrochemical Impedance Spectroscopy during Extended Immersion Tests

EIS measurements are often used to characterize the corrosion protection of coated samples [65]. Nyquist and Bode diagrams obtained after 6, 36 and 96 h are presented in Figure 8. The Nyquist diagrams during first hours exhibit two depressed capacitive loops, typical of metal dissolution and the formation of corrosion product film [66]. For the three configurations, the loop diameters of the capacitive loops remain almost constant during the 36 first hours suggesting a quite constant corrosion rate. This evolution is in good agreement with the *Rp* evolutions presented in Figure 6.

Two capacitive loops are necessary for the fitting of the experimental data using an electrical equivalent circuit corresponding to a porous electrode [67,68] during the first hours of immersion: a middle frequency loop that corresponds to the charge transfer mechanism associated to the coating dissolution and a low frequency loop that corresponds to the formation of the corrosion products. A progressive shift of the characteristic frequency of the loops towards lower frequency is observed, hindering the presence of the middle frequency loop. So, for long immersion, only one capacitive loop was used to correctly fit the experimental data [69].

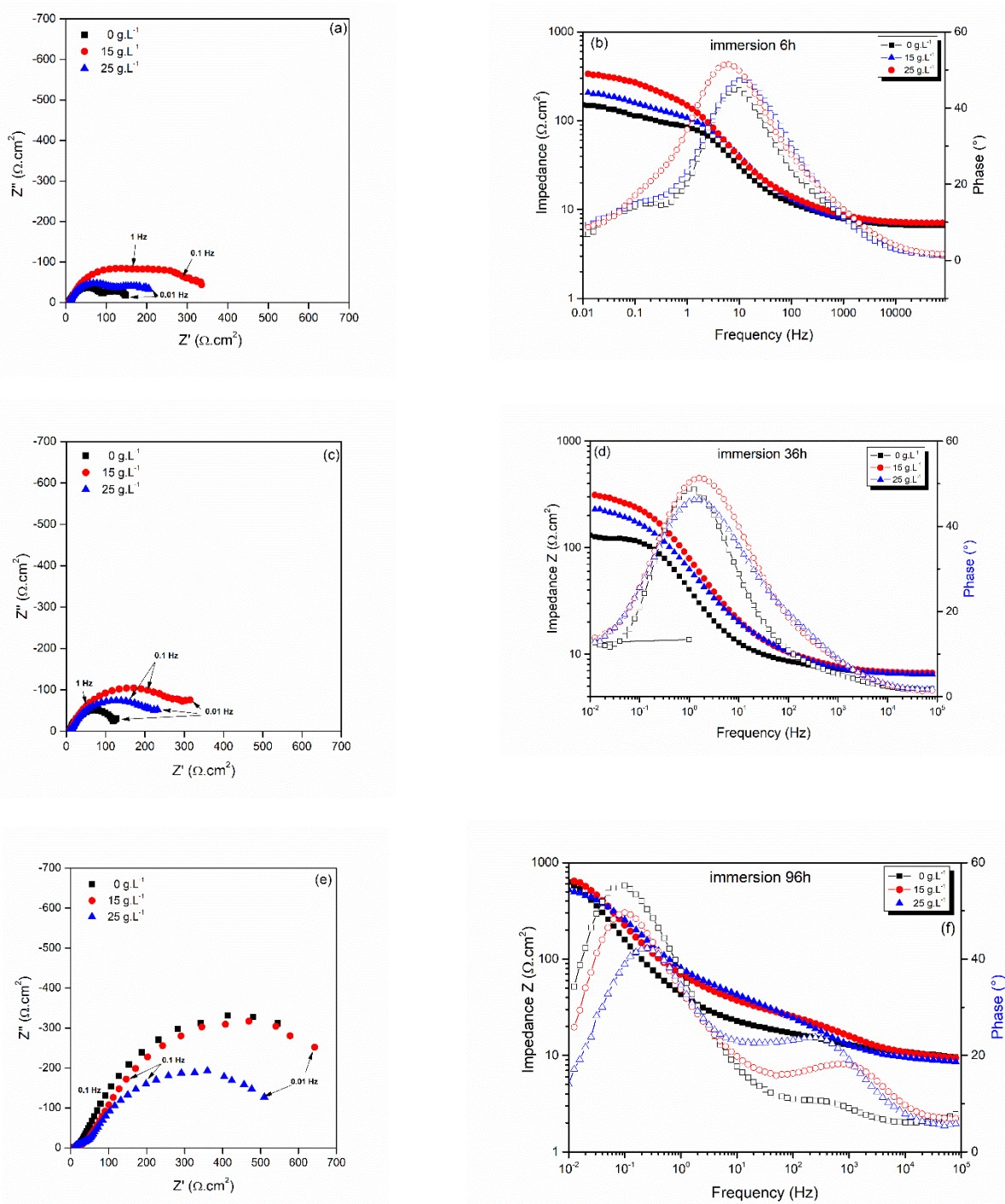

**Figure 8.** Impedance diagrams: Nyquist and Bode plots at different immersion.duration in 35 g·L$^{-1}$ saline solutions: (**a**,**b**) 6 h, (**c**,**d**) 36 h and (**e**,**f**): 96 h.

The 15 g L$^{-1}$ configuration has the largest loop diameter on Nyquist diagram during the first 36 h, underlying a better corrosion resistance and confirming the previously described Rp evolution measurements. It is worth mentioning that electrochemical reactions that eventually might occur at the NP surface are not separated to those corresponding to the zinc coating dissolution. We nevertheless observe a variation in the loop diameter that denotes the impact of the NP presence. The electrochemical parameters deduced from the EIS fitting are listed in Table 3. The formation of the corrosion products leads to the evolution of the parameters, mainly for the constant phase element CPE corresponding

to Q1 and Q2. These parameters are sensitive to the surface roughness evolution and the nature of the zinc corrosion products in saline solution as reported by Anwar et al. [26].

**Table 3.** Electrochemical parameters deduced from the EIS fitting after different durations of immersion in saline solution.

| | Time (h) | Re ($\Omega \cdot cm^2$) | $R_{po}$ ($\Omega \cdot cm^2$) | Q1 ($\Omega^{-1} \cdot cm^{-2} \cdot s^{-n}$) | n1 | Rt ($\Omega \cdot cm^2$) | Q2 ($\Omega^{-1} \cdot cm^{-2} \cdot s^{-n}$) | n2 | C2 (mF·cm$^2$) |
|---|---|---|---|---|---|---|---|---|---|
| Zn-CeO$_2$ 0 g·L$^{-1}$ | 1 h | 8 | 91 | $4.1 \times 10^{-4}$ | 0.91 | 128 | $1.38 \times 10^{-2}$ | 0.5 | 10.1 |
| | 6 h | 8 | 91 | $1.12 \times 10^{-3}$ | 0.81 | 60 | 0.05 | 0.89 | 53.8 |
| | 36 h | 7 | | | | 130 | $5.6 \times 10^{-3}$ | 0.8 | 2.5 |
| | 60 h | 8 | | | | 616 | $7.8 \times 10^{-3}$ | 0.81 | 4.1 |
| | 96 h | 8 | | | | 973 | $8.8 \times 10^{-3}$ | 0.78 | 4.2 |
| Zn-CeO$_2$ 15 g·L$^{-1}$ | 1 h | 8 | 168 | $2.8 \times 10^{-4}$ | 0.78 | 133 | $7.7 \times 10^{-3}$ | 0.5 | 7.1 |
| | 6 h | 7 | 73 | $1.2 \times 10^{-3}$ | 0.99 | 329 | $4. \times 10^{-3}$ | 0.54 | 1.2 |
| | 36 h | 7 | 157 | $2.8 \times 10^{-3}$ | 0.88 | 483 | $1.5 \times 10^{-2}$ | 0.5 | 26.7 |
| | 60 h | 8 | | | | 293 | $5.0 \times 10^{-3}$ | 0.7 | 1.2 |
| | 96 h | 8 | | | | 800 | $6.0 \times 10^{-3}$ | 0.76 | 2.3 |
| Zn-CeO$_2$ 25 g·L$^{-1}$ | 1 h | 8 | 92 | $4.1 \times 10^{-4}$ | 0.81 | 112 | $2 \times 10^{-2}$ | 0.6 | 29.6 |
| | 6 h | 8 | 73 | $4.1 \times 10^{-4}$ | 0.9 | 211 | 0.1 | 0.6 | 542.4 |
| | 36 h | 7 | 96 | $4.1 \times 10^{-4}$ | 0.89 | 166 | $1.0 \times 10^{-2}$ | 0.6 | 6.1 |
| | 60 h | 8 | | | | 357 | $7.4 \times 10^{-3}$ | 0.63 | 1.3 |
| | 96 h | 8 | | | | 600 | $3.6 \times 10^{-5}$ | 0.71 | 0.001 |

Figure 9 presents the evolution of the transfer resistance (Rt), deduced from the fitted experimental values at low frequencies. During the first hours of immersion, Rt evolutions are quite similar to the evolution of the polarization resistance Rp measurement presented in Figure 6. After 50 h of immersion, pure zinc coating presents an increase of Rt probably due to the complete degradation of the coating and the progressive influence of the steel substrate corrosion beneath the zinc corrosion products. We observe that the 15 g L$^{-1}$ sample presents the best corrosion resistance. After 70 h of immersion, a slight Rt increase is observed and denotes a decrease of the coating degradation rate. This beneficial effect of the particle incorporation could be explained by a more compact and dense corrosion product film combined with an inhibiting effect of the homogeneous distributed particles on the coating surface [70]. These observations corroborate results obtained from stationary electrochemical technics. The corrosion product film in presence of ceria nanoparticles is probably more compact, and therefore limiting the infiltration of the electrolyte towards the metallic zinc coating.

### 3.3. Salt Spray Tests

For the different configurations exposed to salt spray test, red rust corresponding to the steel degradation appears after around 158 h. This value is close to Ramanauskas's results obtained with almost similar thickness of zinc coating [71]. Salt spray test, which are widespread in the industrial field [72], seems to be too severe to detect an influence of the ceria nanoparticle incorporation on the corrosion behavior of pure zinc coatings. For all configuration, XRD analysis (Figure 10) reveals the typical sequence of expected zinc corrosion products: firstly, an oxo-hydroxide film is formed and is the basis for a further growth of different compounds such as oxides, carbonates and hydroxides [57].

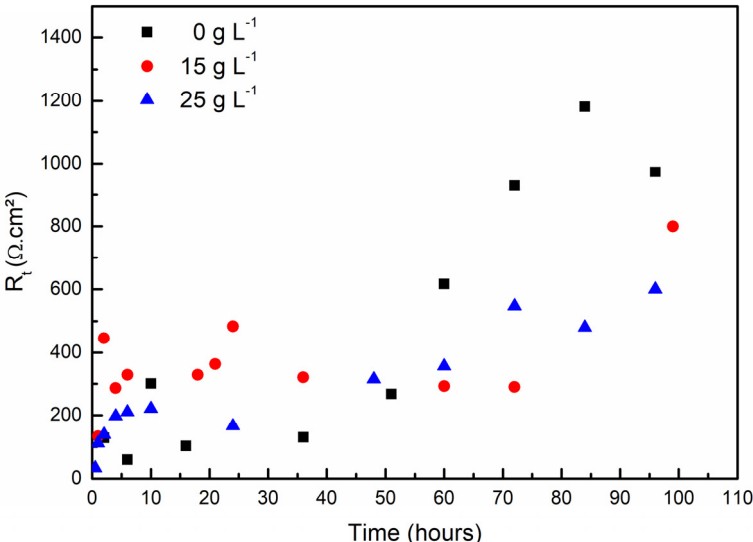

**Figure 9.** Evolution of the transfer resistance at low frequencies versus time for pure zinc, 15 g L$^{-1}$ and 25 g L$^{-1}$ Zn-CeO$_2$ composite coatings electrodeposited on steel substrate, immersed during 96 h in a 35 g L$^{-1}$ saline solution.

The accelerate ageing was stopped after the formation of the first spots of red rust on the surface. The diffraction peaks of the metallic zinc coating are not detected after corrosion. The diffraction peaks of simonkolleite Zn$_5$(OH)$_8$Cl$_2$.H$_2$O (JCPDS 00-007-0155) with the main peak at 2θ = 11° [73] and hydrozincite Zn$_5$(CO$_3$)$_2$(OH)$_6$ (JCPDS n°00-019-1458) with the main peak at around 2θ = 13° are obtained. Diffraction peaks of the steel substrate are observed confirming that the complete dissolution of the metallic zinc matrix during the ageing conditions.

The relative intensities of the zinc corrosion products are quite similar for the 0 and 25 g L$^{-1}$ zinc coatings, with a major contribution of the simonkolleite. According to Yoo et al. [74], the simonkolleite layer improves the corrosion resistance due to its compactness but its stability is affected by the pH and the zinc dissolution rate. For the sample at 15 g L$^{-1}$, it can be observed an increase of the amount of hydrozincite in the corrosion product layer. The composition of the corrosion product layer is changed compared to the other configurations, perhaps in relation with the quite homogeneous dispersion of the ceria nanoparticles in the coatings.

Morphology of the composite coatings is an important key parameter that affects the corrosion protection. A texture along (002) zinc plane is usually associated to a better corrosion behavior [75]. The morphology characterization reveals that the (002) texture is not so marked in the case of our zinc coatings. Surface morphology was quite similar for all the 3 coatings with several orientations of "pack platelets". A columnar structure was observed on the FIB cross-section that could favor the electrolyte infiltration through the pores and intercolumns junctions. As the metallurgical features are quite similar, the amount and dispersion of ceria particles would considerably affect their influence on the corrosion behavior of the composite metal matrix composite. SIMS analyses on composite cross sections reveal that ceria nanoparticles were mainly adsorbed on the surface of the Zn-CeO$_2$ 25 g L$^{-1}$ coating due to the agglomeration phenomenon. So when the surface is degraded during the first hours of salt spray exposure combined with a leaching effect, that is really more important than during the immersion test, the nanoparticles can be detached from the coating surface. The corrosion behavior of this composite becomes then quite similar to that of the as-deposited pure Zn coating.

For the Zn-CeO$_2$ 15 g L$^{-1}$, the nanoparticles are homogeneously distributed through the composite coating thickness and they act differently on the formation of corrosion products, since the amount of hydrozincite is higher inside the corrosion product layer. So

the optimization of the morphology of the zinc coating and a homogeneous distribution of the nanoparticles inside the metal matrix are the key factors that could improve the corrosion resistance.

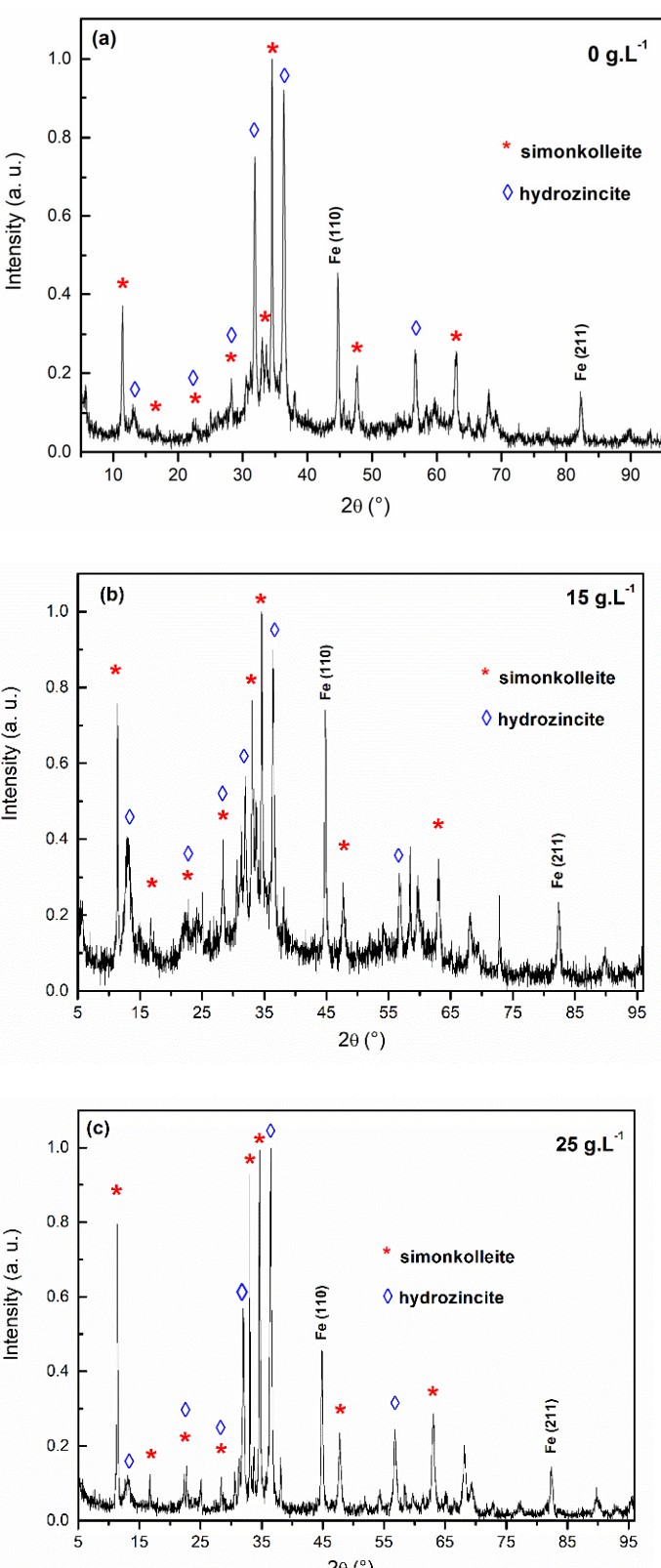

**Figure 10.** X-ray diffractograms of pure zinc and composite coatings after 158 h of salt spray test. (**a**) 0 g L$^{-1}$ (**b**) 15 g L$^{-1}$. (**c**) 25 g L$^{-1}$.

## 4. Conclusions

Optimization of the corrosion behavior of sacrificial zinc coatings is a real challenge. The increase of the nanoparticle concentration in the deposition electrolyte favors the incorporation of particles, but a transition is observed between a well-dispersed particles inside the metallic matrix and a preferential surface adsorption of agglomerated particles. The beneficial effect of incorporated nanoparticles on the corrosion behavior that is reported in the literature due to the reduction of the active surface is partially hindered by the electrochemical activity of the particles at the nanometric scale. Adopting a preliminary metallurgical approach, the role of particles distribution, mostly attached on the edges of the zinc plates and located inside porosities, grain boundaries or triple junction through the coating thickness, could explain the better corrosion behavior during extended immersion tests. The amount of embedded nanoparticles increases with the concentration of nanoparticles in the bath, without modification of the typical zinc coating morphology. We could assume that morphology, which is usually a very important metallurgical parameter that affects the corrosion resistance cannot explain here the slight differences between the both composite deposits. Complementary corrosion tests were performed showing that composite coatings present a better behavior regarding corrosion compared to pure zinc deposit. A shift of corrosion potential for composites was observed during the first hour of immersion, supposing that particles could partially inhibit the corrosion process. Ceria nanoparticles can adsorb oxygen and a competition could take place on the cathodic reaction between the decrease of the dissolved oxygen reduction due to particles screening effect on the coating, and the extraction of the adsorbed oxygen inside the ceria particles. The extraction of adsorbed oxygen on NP surface also provokes the reduction of Ce(IV) in Ce(III), species that normally present a better inhibition effect. A slight improvement of corrosion resistance of the composite deposits, especially for the 15 g L$^{-1}$ configuration, regarding the different corrosion tests was monitored.

Particles, which can act as mechanical support of the growing corrosion products, could therefore protect the weakest parts of the zinc coating, such as cavities and grain boundaries. Our assumption to explain the behavior of 15 and 25 g L$^{-1}$ is that for the last mentioned, particles are more agglomerated, so less homogeneously distributed, and may also introduce more defects in the sacrificial coating during elaboration process. So the optimization of the morphology of the zinc coating and a homogeneous distribution of the nanoparticles inside the metal matrix are the key factors that could improve the corrosion resistance.

**Author Contributions:** Investigation, C.R., R.M. and E.C.; Writing—original draft, L.E.; Project administration, P.S. and J.C. All authors have read and agreed to the published version of the manuscript.

**Funding:** This research was funded by Agence Nationale de la Recherche, grant number 2010 BLAN 939.

**Data Availability Statement:** Not applicable.

**Acknowledgments:** We acknowledge the French Agence National pour la Recherche for supporting this research under the Chameleon Project 2010 BLAN 939. We acknowledge the CLYM federation of microscopy from INSA LYON for the SEM analyses and their support for the FIB-cross section of Zn base composite coatings. We finally acknowledge the Institute Jean LAMOUR from the Lorraine University for their contribution on the GDOES and SIMS analyses of the zinc based coatings.

**Conflicts of Interest:** The authors declare no conflict of interest.

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
