# Peer review of "Corrosion Behavior in Saline Solution of Electrodeposited Nanocomposite Zn-CeO2 Coatings Deposited onto Low Alloyed Steel"

_coatings, doi:10.3390/coatings13101688_

Round 1
Reviewer 1 Report
This manuscript is devoted to studying the Corrosion Behavior in Saline Solution of Electrodeposited Nanocomposite Zn-CeO2 Coatings Deposited onto Low Alloyed Steel. The manuscript is well-written and well organized and includes some merits and can be accepted for publication after incorporating following suggestions:
1. Add quantitative results in abstract.
2. Add some recent literature in introduction.
3. Add novelty of present work in the introduction section.
4. Add supplier details of ceria nanoparticles.
5. Add 2 theta values corresponding to reflection planes (observed in Fig. 2) in section 3.1.
6. “ ……… are involved in the dissolution mechanism of zinc in saline solution….”Add suitable references. (Line 221).
7. “……during the immersion test, the nanoparticles can be detached from the coating surface.” Add suitable references. (Line 446).
8. “For the Zn-CeO2 15 g.L−1 , the nanoparticles are homogeneously distributed through the composite coating thickness….” How did you confirm the homogeneous distribution of nanoparticles? (Line 449).
9. Compare corrosion results of prepared Zn composite coatings with past studies.
Minor editing of English language required
Author Response
REVIEW 1
This manuscript is devoted to studying the Corrosion Behavior in Saline Solution of Electrodeposited Nanocomposite Zn-CeO2 Coatings Deposited onto Low Alloyed Steel. The manuscript is well-written and well organized and includes some merits and can be accepted for publication after incorporating following suggestions:
- Add quantitative results in abstract.
Quantitative results were inserted in the abstract section.
- Add some recent literature in introduction.
New references were added in the introduction
- Add novelty of present work in the introduction section.
The end of the introduction was changed in order to highlight the objective of the article on the comprehension of the influence of particles incorporation on the corrosion behavior
- Add supplier details of ceria nanoparticles.
Ceria particle are from Sigma Aldrich, it is added.
- Add 2 theta values corresponding to reflection planes (observed in Fig. 2) in section 3.1.
Values were added in the text, mainly for the diffraction peak corresponding to the main peak at (002) and (101).
6, 7. …… are involved in the dissolution mechanism of zinc in saline solution….”Add suitable references. (Line 221). And during the immersion test, the nanoparticles can be detached from the coating surface.” Add suitable references. (Line 446).
References were added in the text in order to justify these assumptions.
- “For the Zn-CeO2 15 g.L−1 , the nanoparticles are homogeneously distributed through the composite coating thickness….” How did you confirm the homogeneous distribution of nanoparticles? (Line 449).
Figure 4 was change and SIMS element maps was proposed in order to highlight the homogeneous dispersion of ceria particles inside the metal matrix
- Compare corrosion results of prepared Zn composite coatings with past studies.
Corrosion behavior of Zn composite are compared with the literature.
Reviewer 2 Report
The authors studied Corrosion behavior in saline solution of electrodeposited nanocomposite Zn-CeO2 coatings deposited onto low alloyed steel using electrochemical techniques and calculating some kinetics parameters. The work sounds good and could be published after minor consideration
In paragraph 2.1. Materials and coating elaboration, I think it would be useful to also mention in this paper the of the alkaline galvanic deposition bath composition, used in the experimental studies, as well as the parameters at which the electrodepositions were carried out. Why did you choose 8 μm and not 5 or 10 μm as the optimal thickness for deposits? Please correct in paragraph 25 g.L−1 (the sign ‘.’ must be deleted.
Row 100, please replace was estimated with was calculated.
Rows 120-123 It would be useful to present, along with the description, an image of the equivalent electric circuit.
Row 163, please correct 0.g.L−1 with 0 g L−1
In Table 1. Electrochemical parameters deduced…. the values ​​of the cathodic slopes can also be presented
Rows 235-237 “These evolutions could be associated to the presence of ceria nanoparticles on the coating surface that progressively modify the rate determining step in the global cathodic reaction of dioxygen reduction.” Starting from the explanations given in this paragraph, can the authors offer more explanations regarding the new kinetics of the electrode process, under the conditions in which CeO2 nanoparticles adhere to the steel surface?
Row 277 – For Open circuit potentials (OCP) measured during a 96h immersion test in a 35 g.L−1 NaCl solution, was a single set of determinations made?
In Table 2. Electrochemical parameters deduced from the EIS fitting after different durations of immersion in saline solution, for Zn–CeO2 25 g.L−1 samples: The value 4.1 10−4 constant for Q1 is correct? 542.4 and 0.001 values for Q2 are correct? How many times were these spectra drawn and what is the degree of similarity of the results?
Author Response
REVIEW 2
The authors studied Corrosion behavior in saline solution of electrodeposited nanocomposite Zn-CeO2 coatings deposited onto low alloyed steel using electrochemical techniques and calculating some kinetics parameters. The work sounds good and could be published after minor consideration
- In paragraph 2.1. Materials and coating elaboration, I think it would be useful to also mention in this paper the of the alkaline galvanic deposition bath composition, used in the experimental studies, as well as the parameters at which the electrodepositions were carried out. Why did you choose 8 μm and not 5 or 10 μm as the optimal thickness for deposits? Please correct in paragraph 25 g.L−1(the sign ‘.’ must be deleted.
Description of the deposition process is done. The thickness was deduced from cross-section observations. The duration of the deposition process is maintained constant for all the configuration
- Row 100, please replace was estimated with was calculated
Modification is done.
- Rows 120-123 It would be useful to present, along with the description, an image of the equivalent electric circuit.
The Equivalent electric circuit used for the modeling of the Nyquist and Bode diagrams are conventional. For the initial immersion time, a classic porous electode EE circuit is used, and for extended immersion test, the formation of the corrosion products covers the surface of the sample, that simplifies the equivalent circuit at a randles one. As the EEC are conventional, we only describe the elements that composes the circuit.
- Row 163, please correct 0.g.L−1 with 0 g L−1
Modification is made
- In Table 1. Electrochemical parameters deduced…. the values ​​of the cathodic slopes can also be presented
Cathodic slopes are presented in the table 1 – electrochemical parameters and these values are commented in the text.
- Rows 235-237 “These evolutions could be associated to the presence of ceria nanoparticles on the coating surface that progressively modify the rate determining step in the global cathodic reaction of dioxygen reduction.” Starting from the explanations given in this paragraph, can the authors offer more explanations regarding the new kinetics of the electrode process, under the conditions in which CeO2 nanoparticles adhere to the steel surface?
New comments are added in the text, with the reference explaining the impact of the ceria nanoparticles on the surface reactivity
- Row 277 – For Open circuit potentials (OCP) measured during a 96h immersion test in a 35 g.L−1 NaCl solution, was a single set of determinations made?
No, all the corrosion experiment were made twice for the reproducibility.
- In Table 2. Electrochemical parameters deduced from the EIS fitting after different durations of immersion in saline solution, for Zn–CeO2 25 g.L−1 samples: The value 4.1 10−4 constant for Q1 is correct? 542.4 and 0.001 values for Q2 are correct? How many times were these spectra drawn and what is the degree of similarity of the results?
These values are correct and check twice during these experiments, and take into account the evolution of the surface of the zinc composite matrix with the corrosion products.
Reviewer 3 Report
In this article, the authors aim to deposit Zn-CeO2 nanocomposite coatings were deposited onto mild steel substrates by electrodeposition process and characterize this composite structural and electrochemical property. The topic of this study is novel.
The abstract briefly describes the motivation and results of the study. In the introduction, the authors describe well the problem being solved and the proposed ways for this. All synthesis methods contain a detailed description of the operations performed. The results obtained are presented in the form of figures of good quality and informative. The revealed regularities have an explanation.
I have the following comments/concerns regarding the submitted manuscript that could potentially improve the overall quality and readability of the submitted manuscript.
1. The authors need to clearly highlight their main contribution and novelty of their proposed experimental design over the already reported in the literature.
2. The introduction section is quite elaborate. However, I suggest that the authors add some recent papers related to the recent developments in this field to justify their improved design/experiments over the existing work reported in the recent literature.
Author Response
Reviewer 3
In this article, the authors aim to deposit Zn-CeO2 nanocomposite coatings were deposited onto mild steel substrates by electrodeposition process and characterize this composite structural and electrochemical property. The topic of this study is novel.
The abstract briefly describes the motivation and results of the study. In the introduction, the authors describe well the problem being solved and the proposed ways for this. All synthesis methods contain a detailed description of the operations performed. The results obtained are presented in the form of figures of good quality and informative. The revealed regularities have an explanation.
I have the following comments/concerns regarding the submitted manuscript that could potentially improve the overall quality and readability of the submitted manuscript.
- The authors need to clearly highlight their main contribution and novelty of their proposed experimental design over the already reported in the literature.
The end of the introduction paragraph was changed in order to highlight the objective and the novelty of the research.
- The introduction section is quite elaborate. However, I suggest that the authors add some recent papers related to the recent developments in this field to justify their improved design/experiments over the existing work reported in the recent literature.
New references were added in the introduction section in order to give details and information on the field of the MMC.
Reviewer 4 Report
This article deposited Zn-CeO2 nanocomposite coatings onto mild steel substrates by electro- deposition process. It tried to highlight the effect of ceria nanoparticles embedded into a metallic matrix on the corrosion behavior in saline environment. The corrosion behavior deduced from a multiscale experimental approach (electrochemical measurements as well as neutral salt spray test) was discussed in order to better understand the influence of the nanoparticle incorporation. This work is important and interesting for the readers in steel industry and corrosion resistant field. After answering the following questions, it can be published in coatings:
1)Figs 1 and 2, the particles are CeO2 and Zn is in film shape. The paticles amount is so small, and the XRD can give the involved peak, is it possiple to be repeated?
2)Figs 3 and 4, if authors give the Ce or Zn mapping, it will be better.
3)Figs. 5 and 6, the CeO2 seems unbenefit to the corrosion resistance of the mild steel, why?
4)Fig. 7f, there are two frequency characteristic in the phase angle curves, is it related to the zn or zn-ceo2 coating?
ok
Author Response
Dear reviewer,
Thank you very much for the suggestions and modifications for the article on the corrosion behavior of zinc based composite coatings. You can find the answer and action made in response to the comments.
Best regards
REVIEW 4
This article deposited Zn-CeO2 nanocomposite coatings onto mild steel substrates by electro- deposition process. It tried to highlight the effect of ceria nanoparticles embedded into a metallic matrix on the corrosion behavior in saline environment. The corrosion behavior deduced from a multiscale experimental approach (electrochemical measurements as well as neutral salt spray test) was discussed in order to better understand the influence of the nanoparticle incorporation. This work is important and interesting for the readers in steel industry and corrosion resistant field. After answering the following questions, it can be published in coatings:
1)Figs 1 and 2, the particles are CeO2 and Zn is in film shape. The paticles amount is so small, and the XRD can give the involved peak, is it possiple to be repeated?
When the particle concentration in the bath increase, agglomerate particles are mainly adsorbed at the surface of the coating (for example 25 g L-1), so the presence of these particles is detected on the XRD diagrams (reproducibility was check). For lowest concentration in the bath, the dispersion of the particles is more homogeneous in the film, and the presence of the XRD peak of CeO2 is not clearly detected.
2)Figs 3 and 4, if authors give the Ce or Zn mapping, it will be better.
Figure 4 was changed with a SIMS elements maps in the cross section of the coatings
3)Figs. 5 and 6, the CeO2 seems unbenefit to the corrosion resistance of the mild steel, why?
Ceria Nanoparticles seem to contribute to the cathodic reaction occurring at the zinc surface. The beneficial effect of incorporated nanoparticles on the corrosion behavior that is reported in the literature due to the reduction of the active surface is partially hindered by the electrochemical activity of the particles at the nanometric scale. The beneficial effect is however highlighted for extended immersion test because the particles could play a role on the formation of a more adherent and dense corrosion product film.
4)Fig. 7f, there are two frequency characteristic in the phase angle curves, is it related to the zn or zn-ceo2 coating?
The frequency mentioned in the Nyquist diagrams was related for all the configurations, narrows are present to point the frequency at each curve.
Round 2
Reviewer 2 Report
The authors studied Corrosion behavior in saline solution of electrodeposited nanocomposite Zn-CeO2 coatings deposited onto low alloyed steel
1. I appreciate that the authors reproduced the description of the equivalent electric circuit on page 4 in the dedicated part Corrosion behavior in saline solution, but I suggest that a schematic presentation (image) of the equivalent electric circuit would be welcome, especially since it is easy to export from Z-View.
2. "Potentiodynamic polarization curves were plotted from -150 mV (vs OCP) in the cathodic side up to + 150mV in the anodic side using a sweep rate of 0,2 mV/s:"
- on page 11 the Tafel curves for corrosion processes are shown: why are 2 graphs shown? the only difference in my opinion is the range of potentials that can be found represented on the abscissa.
- where the graph shows the 150 mV polarization in the cathodic or anodic sense, in the cathodic sense, so an interval of 300 mV on which the determinations were made.
-- according to the values ​​of the corrosion currents, adding nanoparticles to the bath has a combined effect, at a concentration of 15 g/L it helps to increase corrosion resistance, after which at 25 g/L the corrosion speed increases to a value greater than that existing in the absence nanoparticles, similarly varying the resistance to polarization. How do the authors explain this behavior? Was it reproducible for multiple sets of samples?
3. OCP evolutions during extended immersion tests suggests, in my opinion, a better corrosion resistance of the sample with the addition of 25 g/L nanoparticles than that of pure zinc, as previously shown by the linear polarization method. (from the Tafel method it is suggested that moving towards more negative values ​​has the effect of increasing corrosion resistance)
Author Response
- I appreciate that the authors reproduced the description of the equivalent electric circuit on page 4 in the dedicated part Corrosion behavior in saline solution, but I suggest that a schematic presentation (image) of the equivalent electric circuit would be welcome, especially since it is easy to export from Z-View.
A figure was added in the procedure part (figure 1) to describe the Equivalent electric circuit used for the EIS.
2 "Potentiodynamic polarization curves were plotted from -150 mV (vs OCP) in the cathodic side up to + 150mV in the anodic side using a sweep rate of 0,2 mV/s:" - on page 11 the Tafel curves for corrosion processes are shown: why are 2 graphs shown? the only difference in my opinion is the range of potentials that can be found represented on the abscissa. - where the graph shows the 150 mV polarization in the cathodic or anodic sense, in the cathodic sense, so an interval of 300 mV on which the determinations were made.
In the figure 6, presenting the polarization curves of the zinc-based coatings, we have limited the current density scale at 0,01 A.cm-2 so the curves were cutted in the anodic part, this is the explanation of the range of potential that was not in perfect agreement with the experimental procedure (300 mV). So we have modified the figure 6 in order to present all the potential domain of the polarization curves
-- according to the values of the corrosion currents, adding nanoparticles to the bath has a combined effect, at a concentration of 15 g/L it helps to increase corrosion resistance, after which at 25 g/L the corrosion speed increases to a value greater than that existing in the absence nanoparticles, similarly varying the resistance to polarization. How do the authors explain this behavior? Was it reproducible for multiple sets of samples?
Thank you for this remark. It is reported in the literature that when the concentration in solution of the particles is high, agglomeration is favored and these agglomerates are preferentially adsorbed on the top surface of the coating. A decrease of the corrosion resistance is then observed. So the improvement of the corrosion resistance depends on the particle concentration in solution and the ability to limit their agglomeration. A paragraph is added in the text.
- OCP evolutions during extended immersion tests suggests, in my opinion, a better corrosion resistance of the sample with the addition of 25 g/L nanoparticles than that of pure zinc, as previously shown by the linear polarization method. (from the Tafel method it is suggested that moving towards more negative values has the effect of increasing corrosion resistance
Thank you for this remark that is very interesting, but as mentioned in figure 7, the Rp values of the sample a 25 g/L are lower than the ones of 15 g/L. The Rp is inversely proportional to the corrosion current density, so we suppose that the localized corrosion induces by the adsorbed agglomerated in the 25g/l sample increases the reactivity of the coating. The OCP, at around 70h of immersion, becomes nobler, suggesting that the degradation of the 25 g/L coating is important, whereas the OCP of the 15 g/L remains quite constant, suggesting that the corrosion resistance is slightly better. Another part that is important, is that the CeO2 nanoparticles seems to interfere with the cathodic reaction of reduction of dioxygen, (evolution of the cathodic Tafel slopes), playing a role in the localized corrosion that could occur at the vicinity of the agglomerates. A uniform distribution on the top surface and/or inside the coating would minimize this localized corrosion and favor the formation of a denser corrosion product film, that is why the 15 g/L sample presents the best corrosion resistance.

Round 3
Reviewer 2 Report
The manuscript has been sufficiently improved to warrant publication in Coatings Journal.